# Comparison of Three Diagnostic Definitions of Metabolic Syndrome and Estimation of Its Prevalence in Mongolia

**DOI:** 10.3390/ijerph20064956

**Published:** 2023-03-11

**Authors:** Enkhtuguldur Myagmar-Ochir, Yasuo Haruyama, Nobuko Takaoka, Kyo Takahashi, Naranjargal Dashdorj, Myagmartseren Dashtseren, Gen Kobashi

**Affiliations:** 1Department of Public Health, School of Medicine, Dokkyo Medical University, 880 Kitakobayashi, Mibu, Shimotsuga-gun, Tochigi 321-0293, Japan; takaoka@dokkyomed.ac.jp (N.T.); k-tak@dokkyomed.ac.jp (K.T.); genkoba@dokkyomed.ac.jp (G.K.); 2Integrated Research Faculty for Advanced Medical Sciences, School of Medicine, Dokkyo Medical University, 880 Kitakobayashi, Mibu, Shimotsuga-gun, Tochigi 321-0293, Japan; yasuo-h@dokkyomed.ac.jp; 3Onom Foundation, Onom Foundation Central Office, 3 Bogd Javzandamba 15 Khoroo, Ulaanbaatar 17011, Mongolia; dashdorj@onomfoundation.org; 4Department of Family Medicine, School of Medicine, Mongolian National University of Medical Sciences, Jamyan Street 3, Ulaanbaatar 14210, Mongolia; myagmartseren@mnums.edu.mn

**Keywords:** agreement, definitions, metabolic syndrome, Mongolia, prevalence

## Abstract

We sought to estimate the prevalence of metabolic syndrome (MS) in the urban population of Mongolia and suggest a preferred definition. This cross-sectional study comprised 2076 representative samples, which were randomly selected to provide blood samples. MS was defined by the National Cholesterol Education Program’s Adults Treatment Panel III (NCEP ATP III), the International Diabetes Federation (IDF), and the Joint Interim Statement (JIS). The Cohen’s kappa coefficient (κ) was analyzed to determine the agreement between the individual MS components using the three definitions. The prevalence of MS in the 2076 samples was 19.4% by NCEP ATP III, 23.6% by IDF, and 25.4% by JIS criteria. For men, moderate agreement was found between the NCEP ATP III and waist circumference (WC) (κ = 0.42), and between the JIS and fasting blood glucose (FBG) (κ = 0.44) and triglycerides (TG) (κ = 0.46). For women, moderate agreement was found between the NCEP ATP III and high-density lipoprotein cholesterol (HDL-C) (κ = 0.43), and between the JIS and HDL-C (κ = 0.43). MS is highly prevalent in the Mongolian urban population. The JIS definition is recommended as the provisional definition.

## 1. Introduction

Metabolic syndrome (MS) is a combination of clustering risk factors, including central obesity, dyslipidemia, hyperglycemia, and hypertension, which eventually lead to cardiovascular diseases (CVDs) and diabetes [1,2,3,4]. The prevalence of MS is increasing worldwide, and many prior studies reported a varying prevalence of MS ranging from 20% to 30% in most countries, depending on the ethnicity, aging, sex, and race of the population [3,4,5,6]. In addition, the prevalence of MS has increased in Asia, including in Mongolia [7,8]. Mongolia has faced increased mortality from non-communicable diseases (NCDs) [9,10], such as CVDs, precipitated by MS. However, there is not yet an established diagnostic definition for MS in the Mongolian population. Therefore, in order to establish adequate criteria, this study compares three diagnostic definitions: the National Cholesterol Education Program Expert Panel on Detection, Evaluation, and Treatment of High Blood Cholesterol in Adults (NCEP ATP III) [1,2]; the International Diabetes Federation (IDF) [3]; and the Joint Interim Statement (JIS) of the International Diabetes Federation Task Force on Epidemiology and Prevention: National Heart, Lung, and Blood Institute; American Heart Association; World Heart Federation; International Atherosclerosis Society; and International Association for the Study of Obesity [4], while adapting them to the Mongolian population.

Previous studies have reported the prevalence of MS in Mongolia. A previous Mongolian epidemiological study of MS reported the prevalence by IDF criteria (32.8%) in adults aged ≥ 40 years [11]. Another comparative study established the MS prevalence by the NCEP ATP III criteria (12–16%) in adults aged over 30 years, which was found to be higher than that in the Japanese and Korean populations [12]. A recent study on Mongolian national trends in MS reported that MS increased significantly (*p* for trend 0.023) when the JIS criteria were followed [13]. These studies have reported basic data on the prevalence of MS. To date, there is scarce information regarding the prevalence of MS in urban adults aged more than 20 years and the comparison of diagnostic definitions, increasing the evidence that early detection and prevention are targeted in the Mongolian urban population.

This seemingly different prevalence appears to be due to the use of the different thresholds and set of criteria established in different definitions, with varying cut-off values for waist circumference (WC), high-density lipoprotein cholesterol (HDL-C), or fasting blood glucose (FBG), and having different ways of combining and including them in blood pressure (BP), triglycerides (TG), and medications for hypertension, diabetes, and dyslipidemia to define MS. The NCEP ATP III definition does not require any specific risks, and it recognizes that MS is a complex disorder [1,2]. Therefore, IDF and JIS definitions use WC cut-off points based on ethnicity [3,4], and WC is now recognized as an important factor in the IDF definition [3]. However, the adaptability of different definitions to different populations is always arguable [3]. With the importance of early screening, determining, and diagnosing MS to prevent mortality from this condition, it is crucial to establish a personalized definition of MS in Mongolia, an Asian country where westernization is increasing.

Therefore, this study compared the differences in MS prevalence among Mongolian urban adults based on three currently used definitions of MS that have their own features to clarify the epidemiological situation of MS, and provide the necessary evidence for preparing diagnostic definitions. Furthermore, we determined the preferred provisional definition of MS for the Mongolians.

## 2. Materials and Methods

### 2.1. Study Design, Sampling, and Population

We conducted a cross-sectional survey on the prevalence of MS in an urban population in Mongolia. The survey followed the guidelines of the World Health Organization (WHO) STEPS Surveillance Manual which provides a complete overview, including guidelines and Appendix A, for countries wishing to undertake NCD risk factor surveys using the WHO STEPwise approach [14], and multistage cluster sampling was conducted for Mongolian residents. First, geopolitical units were sampled and then residents were sampled within these units. In Ulaanbaatar, 142 family healthcare centers (FHCCs) provide primary healthcare services to all citizens. According to the WHO STEPS Surveillance Manual, which recommends that at least 50 primary sampling units (PSUs) be selected from over 100, proportional probability sampling was used to select 52 FHCCs from the eight districts in the first stage of cluster sampling. In the next stage, 88 individuals aged ≥ 20 years were randomly selected from the registers of each of the 52 FHCCs. If the participants could not be reached by the research team, they were replaced by the next participant within the same age and sex category. In total, 4515 urban residents were included in this survey (response rate: 98.7%), and 2258 residents in Mongolia underwent biochemical measurements. A pilot study was conducted on five randomly selected FHCCs in November 2017. Data collection was conducted between December 2017 and January 2018, and the final study population included 2076 urban residents.

We obtained a de-identified dataset of Mongolians from the Onom Foundation, according to the Data Transfer Agreement. Written informed consent was obtained before conducting interviews and physical measurements. The study was approved by the Medical Ethics Committee of the Ministry of Health, Mongolia, and the Ethical Committee of Dokkyo Medical University (Protocol Number, 2021-014).

### 2.2. Measurements

The WHO STEPwise approach is comprised of three steps of risk factor assessment: questionnaire, physical measurements, and biochemical measurements. Before data collection, all field members, who were medical researchers, doctors, nurses, and laboratory technicians, successfully completed 5-day training programs regarding how to conduct interviews, measure anthropometry and BP, and take and collect blood samples. These training programs were organized by the ‘Technical Working Group’ in the Public Health Institute in collaboration with the WHO country office and experts from the relevant cooperating organizations. The pilot study was organized covering all steps of the actual survey. The entire data collection procedure was conducted using an electronic tablet (Fire HD 8, Amazon, Seattle, WA, USA). To avoid data loss, an Android application with an offline mode, QuickTapSurvey (TabbleDabble Inc., Toronto, ON, Canada), was used.

Interviews were conducted using the Mongolian version [15] of the WHO STEPS instrument for NCD risk factor surveillance. To ensure the adequacy of the Mongolian translation of the questionnaires, the Mongolian versions were separately back-translated and reviewed by two independent translators. The survey questionnaire was further adapted to country specifics with the help of local experts, the survey ‘Technical Working Group’, and with close collaboration and technical assistance from the WHO. Finally, it was reviewed and approved by international and national experts and consultants.

Blood pressure measurement: BP was measured using accuracy-validated BP A6 BTs (Microlife Corporation, Taipei, Taiwan) and digital automatic BP monitors. Participants were instructed to abstain from alcohol, cigarette smoking, caffeine consumption, and exercise for at least 30 min before BP measurement. Data collection teams ensured that participants were seated with their legs uncrossed and their back and arm supported, in accordance with the American Heart Association (AHA) guidelines, and that appropriate cuff sizes were used. After a 10 min rest, BP was measured three times, with 3 min intervals between measurements. The average value of the three measurements was then calculated. If one differed by ≥15 mmHg from the other two, it was discarded [16,17].

Physical measurements: Height was measured in centimeters (without shoes) and weight in kilograms (with heavy clothing removed) using a digital scale. Body mass index (BMI) was calculated as weight in kilograms divided by the square of height in meters. We measured the WC of subjects while standing, using a soft tape midway between the lowest rib and iliac crest.

Biochemical measurements: All participants agreed that their blood samples would be collected using a clot activator. The serum samples were centrifuged at 3500 rpm for 10 min. Overnight fasting blood samples were obtained for the measurement of serum lipids and glucose. Concentrations of HDL-C, TG, and FBG following standard operating procedure (SOP)-Liver Center-005 protocols were assessed using a fully automated biochemical analyzer (ERBA-XL200, Mannheim, Germany).

### 2.3. Definition of the Metabolic Syndrome

The NCEP ATP III definition was chosen in this study because Mongolia, as an Asian country, was reported as having a higher BMI compared to Japan and China [18,19]. When using the IDF and JIS definitions, the other two widely accepted definitions of MS chosen in this study, we followed the Asian population thresholds for abdominal obesity [3,4]: WC ≥ 90 cm for men, and ≥80 cm for women. The NCEP ATP III, IDF, and JIS definitions are different from each other in the diagnostic process (Table 1).

### 2.4. Socioeconomic Status (SES) Variables

Education variables were divided into two categories of high school and lower (≤12 years) and higher educational attainment (>12 years) according to the Mongolian education system. Occupational class was defined into three groups: non-manual, manual, and “others” referring to the Erikson–Goldthorpe–Portocarero scheme [20]. Individuals classified as students, retired, and individuals whose stated occupation could not be classified were placed in the “others” group. Monthly income was divided into three types: upper, middle, and lower, according to the average salary per month [21]. The average monthly salary in Mongolian is about MNT 1,500,000 (USD ~450) [22]. Housing was categorized into two types: apartment and Ger district. Ulaanbaatar, the capital city of Mongolia, consists of two different housing-type areas: apartment areas, which are located in the central part of the city; and “Ger areas”, which are a very common housing type among nomads located in the suburbs.

### 2.5. Statistical Analysis

Descriptive analyses were used to report the demographic characteristics of the study participants and the prevalence of MS. The sample was divided into three age groups: young adults (<40), middle adults (40–59), and old adults (60 and over) [23]. For validity between the individual MS components and different definitions, we measured the sensitivity and specificity. Cohen’s kappa coefficient (95% confidence interval (CI)) (poor, κ ≤ 0.20; fair, κ = 0.21–0.40; moderate, κ = 0.41–0.60; substantial, κ = 0.61–0.80; very good, κ > 0.80) was used to determine the level of agreement [24]. As a result of the greater agreement, we used the JIS definition to estimate the odds ratios (ORs) and 95% CIs of SES for MS prevalence using logistic regression. The analyses were stratified according to sex. All *p* values were two-sided, and the alpha level was set at 0.05. Data were analyzed using SPSS version 28.0 (SPSS Inc., Chicago, IL, USA). Mosaic plots were shown for the relationships between MS and individual MS components using JMP Statistical Discovery Software version 16 (SAS Institute Inc., Cary, NC, USA).

## 3. Results

Table 2 shows the characteristics of the study population and the clinical components of MS in Mongolians. Of the total study participants, the mean age was 39.9 ± 13.7 years, and 53.6% were female. The height, weight, WC, BP, and TG levels were all higher in men, whereas the HDL-C levels were lower. The differences in height, weight, WC, and BP between males and females were statistically significant (*p* < 0.05).

The overall prevalence rate of MS was 19.4% according to the NCEP ATP III, 23.6% according to the IDF, and 25.4% according to the JIS definitions. The prevalence of MS was higher in female participants and in the 40–59 and 60 and over age groups when stratified by sex and age (Figure 1).

The agreements between the NCEP ATP III, IDF, and JIS definitions, and the individual MS components are presented in Table 3. For men, a moderate agreement was found between the NCEP ATP III and WC (κ = 0.42), and between the JIS and FBG (κ = 0.44) and TG (κ = 0.46). For women, a moderate agreement was found between the NCEP ATP III and HDL-C (κ = 0.43), and between the JIS and HDL-C (κ = 0.43).

Table 4 shows the sex-divided ORs of SES factors for MS prevalence based on the JIS definition, which had moderate agreement between more MS components in men and showed a higher prevalence in both sexes. In men, the 40–59, and 60 and over age groups and married participants were significantly associated with MS, while for women, the 40–59, and 60 and over age groups as well as those with lower educational attainment, “others” group for occupation, and married participants were significantly associated with MS. In the multivariable logistic regression analysis, the 40–59 age group (aOR = 1.44, 95% CI 1.01 to 2.04 in men, aOR = 2.28, 95% CI 1.68 to 3.09 in women) and female 60 and over age group (aOR = 3.22, 95% CI 2.04 to 5.08) in addition to married female participants (aOR = 1.57, 95% CI 1.04 to 2.36) were significantly associated with MS.

Mosaic plots showed the MS distribution within the individual MS components by sex. Male participants with elevated MS components had a higher percentage of MS based on the JIS definition except for WC. For females, high glucose levels, and elevated cholesterol level participants had a higher percentage of MS according to the JIS definition. Based on the IDF definition, the elevated systolic blood pressure (SBP) participants had a higher percentage of MS, and the elevated diastolic blood pressure (DBP) participants had a higher percentage of MS based on both JIS and IDF definitions. However, according to the NCEP ATP III definition, the high WC participants had a higher percentage of MS (Appendix A).

## 4. Discussion

Using an urban population in Mongolia, we established the prevalence of MS, as 15.5%, 21.6%, and 23.8%, respectively, in men, and 22.8%, 25.3%, and 26.8%, respectively, in women as defined by the NCEP ATP III, IDF, and JIS definitions. Concerning the level of agreement, JIS definition had moderate agreements with more MS components in men, while both the NCEP ATP III and JIS definition had moderate agreements with HDL-C in women. According to our knowledge, this is the first study determining the preferred provisional definition of MS for the Mongolians.

Considering the greater emphasis of the JIS definition on FBG and TG for Mongolian men, and on HDL-C for Mongolian women, a moderate agreement between the JIS and these components is plausible. Furthermore, the agreement of the JIS definition on FBG and TG was slightly higher in women compared to the agreements of the other two definitions. However, the agreement of the JIS definition on WC was lower in both sexes than the agreements of the other two definitions because WC suffers from a higher measurement error [25,26]. Regarding the high prevalence of CVDs among the Mongolian population, applying these criteria to identify persons at risk might be helpful. In addition, the prevalence of MS measured using the JIS definition was higher in both sexes than that measured using other definitions. The Mongolian government has been building policies to prevent and control NCDs [27] and to set the Mongolia Sustainable Development Vision 2030 report, which aims to reduce the main NCDs, such as CVDs, health risk factors, and preventable deaths [28]. For policy responses, addressing MS is necessary. According to the national NCD STEP surveys, the prevalence of most NCD risk factors remained stable, while that of overweight and obesity increased [29,30]. Thus, the JIS definition could be used to identify more people at high risk and make early interventions for controlling MS in urban populations. A moderate agreement was more related to elevated glucose and cholesterol levels. In Mongolia, the incidence of diabetes and dyslipidemia has increased and is more common in the urban areas [29,31]. A cross-sectional national survey reported in 2019 that the prevalence of dyslipidemia was 58.6%, among which 6.2% were aware, 18.9% were treated, and 21.5% were controlled [31]. Another study noted that only a small proportion of the total hypertensive or diabetic population had adequately controlled blood pressure or blood sugar due to a large diagnosis gap, non-treatment of previously diagnosed populations, and inadequate control of the treated population [32].

Traditional and ecological aspects may play a role in the observed patterns of the results. A harsh climate, average atmospheric temperature, atmospheric pressure, precipitation, and mineralization of rivers can have an influence [33]. Traditionally, Mongolians have a unique nomadic lifestyle, and its population prefers a diet of meat, milk, and its derivatives [34,35,36]. Most rural families are physically active, involved in caring for livestock, transportation by horses and camels, milking, shearing, and combing. This unique lifestyle might be associated with a low BMI in Mongolians [30]. However, nearly 67% of the population is living in the capital city, seeking education and a professional job over traditional nomadic life, and migration from rural areas to Ulaanbaatar has been increasing in recent years [37]. For herders, there is a lack of access to education and healthcare services because of nomadic communities living and moving in remote areas [38]. Along with this rapid modernization of the whole country, diet and nutrition have shifted toward a diet with high fat, high energy, and low dietary fibers [39]. Dietary habits are associated with lifestyle-related diseases and early aging in Mongolia [40]. According to the NCD risk factor surveys, urban residents had higher risks, such as smoking, physical inactivity, obesity, and high cholesterol levels [29]. There was a significant association between MS and poor lifestyle habits and some SES factors such as education and marital status [11,13]. Additionally, Mongolian urban adults have a higher prevalence of MS [9,29]. Establishing health promotion policies based on the needs and convenience of urban residents is an urgent matter.

In addition, results of the logistic regression showed that the middle (40–59) and older adult (60 and over) groups had one to three times higher prevalence of MS, and married female participants had a higher prevalence of MS according to the JIS definition, which was suggested to be superior in defining MS in Mongolians than the other two definitions based on our main results. The major role of age in predicting MS was consistent with other previous studies [41]. The present study reported that married women had a higher prevalence of MS than those who were never married, which was consistent with the results of some studies [42]. In our study, unmarried participants were younger (93.1% of unmarried male young adults (<40) and 83.9% of unmarried female young adults (<40)). Nevertheless, due to their traditional social roles, Mongolian married women raise their children, do the housework, and cook the food. This may be related to the higher prevalence of MS components, such as physical inactivity, overweight, and obesity, in Mongolian women when compared to men [29]. Further research on how marital status, especially among women, is associated with MS is needed to clarify these findings.

Several limitations of this study should be noted. First, our cross-sectional study did not allow us to derive conclusions regarding causal mechanisms. Second, we did not evaluate visceral fat using computed tomography (CT) and magnetic resonance imaging (MRI) to measure central obesity. Third, our data were extracted only from the Mongolian urban population using multistage cluster sampling which included some sampling errors. However, our sampling was carried out following the WHO STEPS Surveillance Manual, and selection bias was smaller than non-random sampling. Furthermore, the current study provides the first report of a comparative study of MS prevalence in Mongolians according to three different definitions and suggests a provisional definition for Mongolians.

## 5. Conclusions

In conclusion, the current findings indicate that the prevalence of MS is high among the general population in the urban area of Mongolia. The JIS definition is recommended as the preferred provisional definition to identify more people at risk of MS. Hence, national preventive strategies and interventions should interfere as early as possible in detecting relevant risk factors. Therefore, further discussion is needed to establish the standardized measurement criteria and definitions of MS to allow Mongolians to define cut-offs as the best indicators of morbidity.

## Figures and Tables

**Figure 1 ijerph-20-04956-f001:**
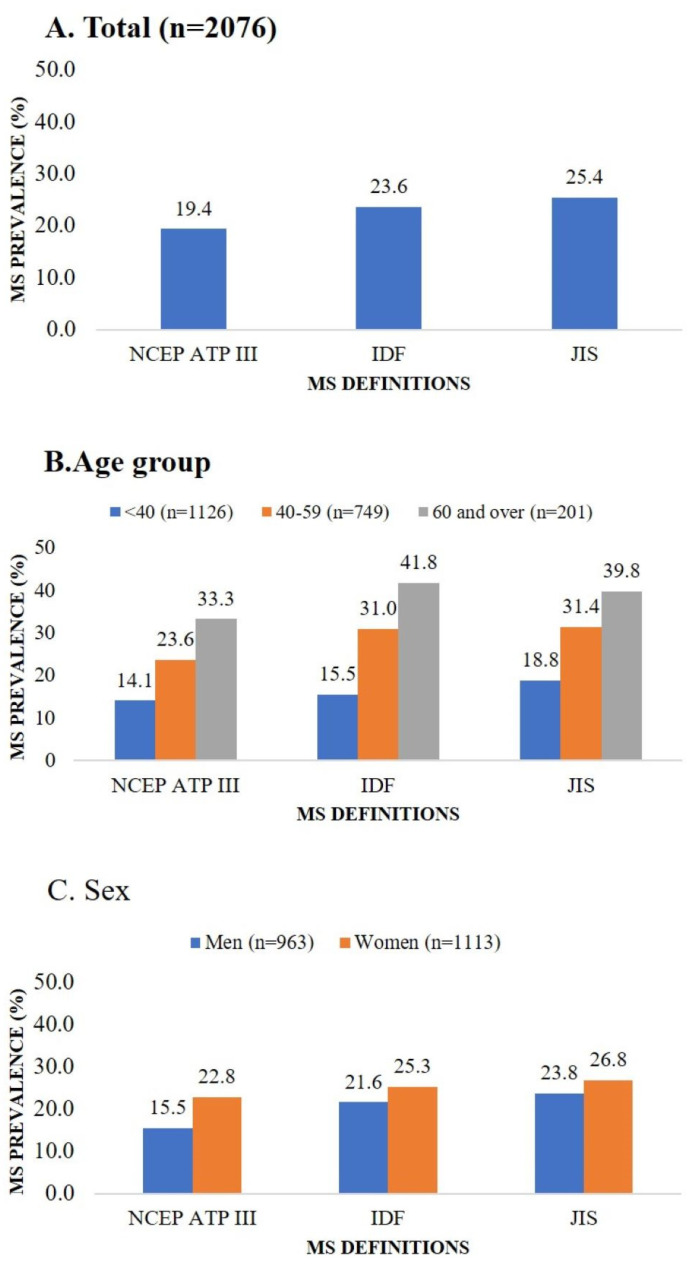
Metabolic syndrome prevalence according to three definitions by total population, age group, and sex. Abbreviations: IDF, International Diabetes Federation; JIS, Joint Interim Statement; MS, metabolic syndrome; NCEP ATP III, National Cholesterol Education Program’s Adults Treatment Panel III.

**Table 1 ijerph-20-04956-t001:** Diagnosis criteria of metabolic syndrome used in the current study.

Risk Factors (RF)	NCEP ATP III (2004)	IDF (2005)	JIS (2009)
Waist circumference (WC)	M ≥ 102 cm F ≥ 88 cm	M ≥ 90 cm F ≥ 80 cm (South Asian cut-points)	M ≥ 90 cm F ≥ 80 cm (South Asian cut-points)
Blood pressure (BP)	SBP ≥ 130 or DBP ≥ 85 mmHg or on treatment for HPT	SBP ≥ 130 or DBP ≥ 85 mmHg or on treatment for HPT	SBP ≥ 130 or DBP ≥ 85 mmHg or on treatment for HPT
Fasting blood glucose (FBG)	≥100 mg/dL or on treatment for elevated glucose	≥100 mg/dL or previously diagnosed T2DM	≥100 mg/dL or on treatment for elevated glucose
Triglycerides (TG)	≥150 mg/dL or on treatment for TG	≥150 mg/dL or on treatment for TG	≥150 mg/dL or on treatment for TG
HDL-C	M < 40 mg/dL F < 50 mg/dL or on treatment for HDL-C	M < 40 mg/dL F < 50 mg/dL or on treatment for HDL-C	M < 40 mg/dL F < 50 mg/dL or on treatment for HDL-C
Metabolic syndrome definitions	Any 3 RF or more	WC+ any 2 RF or more	Any 3 RF or more

DBP, diastolic blood pressure; F, female; HDL-C, high-density lipoprotein cholesterol; HPT, hypertension; IDF, International Diabetes Federation; JIS, Joint Interim Statement; M, male; NCEP ATP III, National Cholesterol Education Program’s Adults Treatment Panel III; RF, risk factors; SBP, systolic blood pressure; T2DM, type 2 diabetes mellitus.

**Table 2 ijerph-20-04956-t002:** Anthropometric and clinical characteristics of the survey participants.

Characteristics	Total	Men	Women	*p* Value
	*n* = 2076	*n* = 963	*n* = 1113	
Age, mean (SD), yr	39.9 (13.7)	39.6 (13.8)	40.2 (13.6)	0.860
Height, mean (SD), cm	164.0 (8.6)	169.9 (7.0)	158.8 (6.2)	<0.001
Weight, mean (SD), kg	70.4 (14.6)	75.9 (14.5)	65.7 (13.1)	<0.001
Body mass index, mean (SD)	26.1 (4.9)	26.2 (4.6)	26.0 (5.1)	0.323
Waist circumference, mean (SD), cm	88.0 (13.4)	90.9 (13.4)	85.4 (12.9)	<0.001
Systolic blood pressure, mean (SD), mmHg	123.4 (17.1)	128.5 (16.1)	119.0 (16.7)	<0.001
Diastolic blood pressure, mean (SD), mmHg	79.0 (11.4)	82.4 (11.0)	76.1 (11.0)	<0.001
HDL-C, mean (SD), mg/dL	58.0 (11.8)	57.8 (11.8)	58.1 (11.9)	0.218
Triglycerides, mean (SD), mg/dL	135.0 (110.1)	136.7(115.9)	133.5 (104.8)	0.308
Fasting blood glucose, mean (SD), mg/dL	95.4 (31.9)	95.3 (31.1)	95.4 (32.5)	0.767
Antihypertensive drug, *n* (%)	427 (20.6)	143 (14.8)	284 (25.5)	<0.001
Glucose drug, *n* (%)	39 (1.9)	14 (1.5)	25 (2.2)	0.185
Cholesterol drug, *n* (%)	86 (4.1)	27 (2.8)	59 (5.3)	0.004

HDL-C, high-density lipoprotein cholesterol; SD, standard deviation.

**Table 3 ijerph-20-04956-t003:** Validity and agreement between individual metabolic syndrome components and three definitions.

Components	Diagnostic Criteria	Sensitivity (%)	Specificity (%)	Kappa Coefficient (95% CI)	Agreement
**Men**					
WC	NCEP ATP III	46	93	0.42 (0.35 to 0.49)	Moderate
	IDF	41	100	0.40 (0.36 to 0.45)	Fair
	JIS	38	92	0.29 (0.24 to 0.34)	Fair
SBP	NCEP ATP III	27	91	0.20 (0.15 to 0.26)	Poor
	IDF	38	88	0.28 (0.22 to 0.34)	Fair
	JIS	38	85	0.26 (0.20 to 0.32)	Fair
DBP	NCEP ATP III	26	91	0.19 (0.13 to 0.25)	Poor
	IDF	36	87	0.24 (0.18 to 0.31)	Fair
	JIS	37	84	0.22 (0.16 to 0.28)	Fair
FBG	NCEP ATP III	38	92	0.34 (0.27 to 0.41)	Fair
	IDF	44	86	0.31 (0.24 to 0.38)	Fair
	JIS	56	87	0.44 (0.37 to 0.50)	Moderate
TG	NCEP ATP III	39	95	0.40 (0.35 to 0.48)	Fair
	IDF	43	88	0.33 (0.27 to 0.40)	Fair
	JIS	54	89	0.46 (0.40 to 0.52)	Moderate
HDL-C	NCEP ATP III	70	88	0.30 (0.21 to 0.38)	Fair
	IDF	52	80	0.13 (0.07 to 0.19)	Poor
	JIS	80	79	0.22 (0.16 to 0.28)	Fair
**Women**					
WC	NCEP ATP III	41	90	0.33 (0.28 to 0.38)	Fair
	IDF	38	100	0.30 (0.26 to 0.33)	Fair
	JIS	36	91	0.21 (0.18 to 0.25)	Fair
SBP	NCEP ATP III	49	84	0.31 (0.25 to 0.38)	Fair
	IDF	54	82	0.34 (0.27 to 0.40)	Fair
	JIS	54	80	0.31 (0.25 to 0.37)	Fair
DBP	NCEP ATP III	47	83	0.28 (0.21 to 0.34)	Fair
	IDF	51	81	0.29 (0.22 to 0.35)	Fair
	JIS	52	79	0.27 (0.21 to 0.33)	Fair
FBG	NCEP ATP III	47	86	0.35 (0.28 to 0.41)	Fair
	IDF	46	82	0.29 (0.22 to 0.35)	Fair
	JIS	55	83	0.39 (0.33 to 0.45)	Fair
TG	NCEP ATP III	47	87	0.36 (0.30 to 0.42)	Fair
	IDF	47	83	0.31 (0.25 to 0.38)	Fair
	JIS	55	84	0.40 (0.34 to 0.46)	Fair
HDL-C	NCEP ATP III	58	87	0.43 (0.37 to 0.50)	Moderate
	IDF	53	82	0.33 (0.27 to 0.40)	Fair
	JIS	64	83	0.43 (0.37 to 0.49)	Moderate

DBP, diastolic blood pressure; FBG, fasting blood glucose; HDL-C, high-density lipoprotein cholesterol; IDF, International Diabetes Federation; JIS, Joint Interim Statement; NCEP ATP III, National Cholesterol Education Program’s Adults Treatment Panel III; SBP, systolic blood pressure; TG, triglycerides; WC, waist circumference.

**Table 4 ijerph-20-04956-t004:** Prevalence and odds ratios for the metabolic syndrome by JIS definition against SES factors.

Characteristics		Men *n* = 963	
	N (%)	Crude OR (95% CI)	Adjusted OR ^a^ (95% CI)
Age group			
<40	527 (54.7)	1.00	1.00
40–59	350 (36.3)	1.59 (1.17 to 2.17)	1.44 (1.01 to 2.04)
60 and over	86 (8.9)	1.77 (1.08 to 2.91)	1.55 (0.87 to 2.78)
Education			
Higher educational attainment	442 (45.9)	1.00	1.00
High school and lower	521 (54.1)	0.99 (0.74 to 1.32)	0.92 (0.65 to 1.30)
Occupation			
Non-manual	224 (25.3)	1.00	1.00
Manual	407 (42.3)	1.13 (0.78 to 1.63)	1.09 (0.72 to 1.66)
Others/students, retired, unemployed	312 (32.4)	1.16 (0.79 to 1.71)	1.08 (0.68 to 1.73)
Marital status			
Unmarried	732 (76.0)	1.00	1.00
Married	231 (24.0)	1.69 (1.17 to 2.44)	1.38 (0.91 to 2.09)
Monthly income			
Upper	243 (25.2)	1.00	1.00
Middle	295 (30.6)	1.14 (0.77 to 1.67)	1.17 (0.79 to 1.74)
Lower	425 (44.1)	1.03 (0.71 to 1.48)	1.03 (0.70 to 1.51)
Housing			
Apartment	481 (49.9)	1.00	1.00
Ger district	482 (50.1)	0.92 (0.69 to 1.23)	0.96 (0.69 to 1.32)
	Women *n* = 1113		
Age group			
<40	599 (53.8)	1.00	1.00
40–59	399 (35.8)	2.51 (1.88 to 3.35)	2.28 (1.68 to 3.09)
60 and over	115 (10.3)	3.89 (2.56 to 5.93)	3.22 (2.04 to 5.08)
Education			
Higher educational attainment	606 (54.4)	1.00	1.00
High school and lower	507 (45.6)	1.33 (1.02 to 1.73)	1.09 (0.80 to 1.47)
Occupation			
Non-manual	336 (30.2)	1.00	1.00
Manual	231 (20.8)	1.16 (0.79 to 1.70)	0.99 (0.65 to 1.51)
Others/students, retired, unemployed	546 (49.1)	1.39 (1.02 to 1.89)	1.12 (0.79 to 1.60)
Marital status			
Unmarried	896 (80.5)	1.00	1.00
Married	217 (19.5)	2.22 (1.51 to 3.25)	1.57 (1.04 to 2.36)
Monthly income			
Upper	200 (18.0)	1.00	1.00
Middle	330 (29.6)	1.01 (0.68 to 1.50)	0.95 (0.63 to 1.45)
Lower	583 (52.4)	1.13 (0.79 to 1.63)	1.02 (0.68 to 1.52)
Housing			
Apartment	544 (48.9)	1.00	1.00
Ger district	569 (51.1)	1.03 (0.79 to 1.34)	1.04 (0.78 to 1.39)

CI, confidence interval; JIS, Joint Interim Statement; OR, odds ratio; SES, socio-economic status; a, using the multivariable logistic regression analysis.

## Data Availability

The data presented in this study are available on request from the corresponding author.

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
