# Peer review of "Comparison of Three Diagnostic Definitions of Metabolic Syndrome and Estimation of Its Prevalence in Mongolia"

_ijerph, 2023, doi:10.3390/ijerph20064956_

Round 1
Reviewer 1 Report
Dear Authors,
Thank you for the opportunity to review your manuscript. Current research focuses on comparing three diagnostic definitions of metabolic syndrome and estimating prevalence of metabolic syndrome in Mongolian population. Though, the research question is interesting, there are some issues with the methodology and analyses, which certainly deserves authors’ attention. My major and minor comments on the manuscript are appended below.
1. Abstract: Please replace the word “superior” with “preferred”
2. There is discrepancy in the study sample mentioned (4,515) in the abstract and under the material methods section (2076). Since the final sample included people with blood sample collected to define the metabolic syndrome (especially for FBG, TG, HDL-C) hence the final number should be included as a sample. Moreover, tables and data analyses also include 2076 as a sample.
3. The abbreviations needs to be expanded wherever they are appearing for the first time and not randomly through the entire manuscript (MS, NCEP ATP III, IDF, JSF, WC under introduction just to list few).
4. Material and Methods: It is not clear in the manuscript if any measures were taken while analyzing the data considering the use multistage cluster sampling which is often associated with high sampling errors and high degree of subjectivity?
5. Furthermore, it is not clear in the manuscript, how data collectors were trained? what method was used for their training? how many of them were involved in data collection? who trained them and their credentials? how interviews were conducted? where were they conducted? were these structured or unstructured etc.?
6. Authors should also provide some greater details about their Mongolian version of the WHO Steps instrument. How was its reliability and validity assessed? How was it translated in Mongolian? Who translated it and their credentials? What method or guideline was used for its cultural and linguistic adaptation and validation?
7. Why did authors chose to include NCEP-ATP-III definition for comparison, even though it is not appropriate to be used in Asian population for measuring waist circumference? And among three different definitions used, all of them are identical on all measures with the exception of WC only in NCEP-ATP-III. Since, the other two IDF and JSS definitions are identical, what was authors' expectation about the disagreement between these two definitions in their sample?
9. What is considered to be an average salary per month in Mongolia? just to make it clear for our readers in context of the world.
9. Please replace the word “background” with “demographic” - characteristics.
10. Please provide foot note below the tables to expand abbreviations (Fr e.g., RF?)
11. Results: Why authors choose to report HDL-C, TG and FBG mean values in mmol/L in Table 2 when they were defined as mg/dl in Table 1. Please be consistent in reporting.
12. How will authors explain the discrepancy in MS prevalence according to IDF (24.8%) and JIS (26.8%) definition, when both definitions are identical for all the individual components?
13. Similar questions can also be raised for the sensitivity, specificity when comparing IDF with JIS definitions and the level of agreement?
14. The sentence discussing multivariate regression analysis is somewhat ambiguous and need to be rephrased (line 203-205)
15. Discussion: The paragraph discussing health literacy is irrelevant and needs to be removed, since health literacy was not assessed in this research and seems like authors conjecture at this point.
16. Authors should expand their limitations considering sampling method, similarities in three definitions etc.
Author Response
Response to Reviewer 1 comments
Thank you for the opportunity to review your manuscript. Current research focuses on comparing three diagnostic definitions of metabolic syndrome and estimating prevalence of metabolic syndrome in Mongolian population. Though, the research question is interesting, there are some issues with the methodology and analyses, which certainly deserves authors’ attention. My major and minor comments on the manuscript are appended below.
We thank the reviewer for these comments. We largely agree with the points raised and considered them in the revised version of the manuscript. Our point-by-point responses to each of the comments are as follows:
Point 1: Abstract: Please replace the word “superior” with “preferred”
Response 1: Thank you for your comment. We have replaced the word “superior” with “preferred” in the Abstract.
Point 2: There is discrepancy in the study sample mentioned (4,515) in the abstract and under the material methods section (2076). Since the final sample included people with blood sample collected to define the metabolic syndrome (especially for FBG, TG, HDL-C) hence the final number should be included as a sample. Moreover, tables and data analyses also include 2076 as a sample.
Response 2: Thank you for your important comment. We have revised the final number of samples as 2076 in the Abstract.
Point 3: The abbreviations needs to be expanded wherever they are appearing for the first time and not randomly through the entire manuscript (MS, NCEP ATP III, IDF, JSF, WC under introduction just to list few).
Response 3: Thank you for your comment. We have expanded the all abbreviations in the main text.
Point 4: Material and Methods: It is not clear in the manuscript if any measures were taken while analyzing the data considering the use multistage cluster sampling which is often associated with high sampling errors and high degree of subjectivity?
Response 4: Thank you for your comment. We have added a limitation of sampling errors in discussion section. We have revised this section accordingly, as follows:
(Original): Several limitations of this study should be noted. First, our cross-sectional study did not allow to derive conclusions regarding causal mechanisms. Second, we did not evaluate visceral fat using computed tomography (CT) and magnetic resonance imaging (MRI) to measure central obesity. Third, our data were extracted only from the Mongolian urban population. However, sampling was carried out following the WHO STEPs Surveillance Manual, and selection bias was small. Despite these limitations, the current study provides the first report of a comparative study of MS prevalence in Mongolians according to three different definitions, and suggests a provisional definition for Mongolians.
(Revised, revised parts are underlined): “Several limitations of this study should be noted. First, our cross-sectional study did not allow to derive conclusions regarding causal mechanisms. Second, we did not evaluate visceral fat using computed tomography (CT) and magnetic resonance imaging (MRI) to measure central obesity. Third, our data were extracted only from the Mongolian urban population using multistage cluster sampling which included some sampling errors. However, our sampling was carried out following the WHO STEPs Surveillance Manual, and selection bias was smaller than non-random sampling. Besides, the current study provides the first report of a comparative study of MS prevalence in Mongolians according to three different definitions, and suggests a provisional definition for Mongolians.”
Point 5: Furthermore, it is not clear in the manuscript, how data collectors were trained? what method was used for their training? how many of them were involved in data collection? who trained them and their credentials? how interviews were conducted? where were they conducted? were these structured or unstructured etc.?
Response 5: Thank you for your important comment. The field members’ training was organized by the ‘Technical Working Group’ in the Public Health Institute in collaboration with the WHO country office and experts from the relevant cooperating organizations. The sixty field members who are medical researchers, doctors, nurses and technicians were trained on the survey methodology, tools and instruments to be used. After the 5-day training, the pilot study was performed. The actual survey was conducted by 11 teams, each with 5-6 members. The residents were announced to come to selected Family Health Centers. Each member in the team performed the interview, anthropometric measurement, and collection of blood samples. There were structured interviews. We have revised this section accordingly, as follows:
(Original): Before data collection, all field members successfully completed training programs regarding conducting interviews, measuring the BP, taking and collecting blood samples.
(Revised, revised parts are underlined): “The WHO STEPwise approach is comprised of three steps of risk factor assessment: questionnaire, physical measurements and biochemical measurements. Before data collection, all field members, who are medical researchers, doctors, nurses and laboratory technicians successfully completed 5-day training programs regarding how to conduct interviews, measure anthropometry and BP, and take and collect blood samples. These training programs were organized by the ‘Technical Working Group’ in the Public Health Institute in collaboration with the WHO country office and experts from the relevant cooperating organizations. The pilot study was organized covering all steps of the actual survey.”
Point 6: Authors should also provide some greater details about their Mongolian version of the WHO Steps instrument. How was its reliability and validity assessed? How was it translated in Mongolian? Who translated it and their credentials? What method or guideline was used for its cultural and linguistic adaptation and validation?
Response 6: Thank you for your comment. The prevalence of MS among the Mongolian population was studied using the 3-step approach recommended by the WHO: Step 1: Questionnaire survey, Step 2: Physical and anthropometric measurements and Step 3: Biochemical measurements.
Based on the revised version of WHO STEPS survey protocol and existing guidelines, the survey scope, instrument and data collection methods were revised to take into consideration the country situation, specifics, needs and capacity. The survey questionnaire was further adapted to country specifics with the help of local experts, the survey ‘Technical Working Group’ and with close collaboration and technical assistance from WHO. Finally, it was reviewed and approved by international and national experts and consultants. To ensure the adequacy of the Mongolian translation of the questionnaires, the Mongolian versions were separately back translated and reviewed by two independent translators and we mentioned it in our main text. The questionnaire consists of 11 main sections including a total of 185 questions. We used only the first section titled “General Information and Demographics”. Also, a previous study used the Mongolian version of this questionnaire and they reported the prevalence of hypertension. (Harry Potts BA et al. 2019) We have revised this section accordingly, as follows:
(Original): Interviews were conducted using the Mongolian version of the WHO STEPS instrument for NCDs Risk Factor surveillance. To ensure the adequacy of the Mongolian translation of the questionnaires, the Mongolian versions were separately back translated and reviewed by two independent translators.
(Revised, revised parts are underlined): “Interviews were conducted using the Mongolian version[15] of the WHO STEPS instrument for NCDs Risk Factor surveillance. To ensure the adequacy of the Mongolian translation of the questionnaires, the Mongolian versions were separately back translated and reviewed by two independent translators. The survey questionnaire was further adapted to country specifics with the help of local experts, the survey ‘Technical Working Group’ and with close collaboration and technical assistance from WHO. Finally, it was reviewed and approved by international and national experts and consultants.”
Point 7: Why did authors chose to include NCEP-ATP-III definition for comparison, even though it is not appropriate to be used in Asian population for measuring waist circumference? And among three different definitions used, all of them are identical on all measures with the exception of WC only in NCEP-ATP-III. Since, the other two IDF and JSS definitions are identical, what was authors' expectation about the disagreement between these two definitions in their sample?
Response 7: We thank the reviewer for this comment. The NCEP ATP III definition was chosen in this study because Mongolia, as an Asian country, was reported as having a higher BMI compared to Japan and China (Shiwaku k et al. 2004; Sun Z et al. 2008). When using the IDF and JIS definitions, the other two widely accepted definitions of MS chosen in this study, we followed the Asian population thresholds for abdominal obesity[3,4]: WC ≥90 cm for men, and ≥80 cm for women. The NCEP ATP III, IDF and JIS definitions are different from each other in diagnostic process. We clarified the main text as follows:
(Original): Three widely accepted definitions of MS used in this study were established by the National Cholesterol Education Program Expert Panel on Detection, Evaluation, and Treatment of High Blood Cholesterol in Adults (NCEP ATP III), International Diabetes Federation (IDF) definition, and Joint Interim Statement of the International Diabetes Federation Task Force on Epidemiology and Prevention: National Heart, Lung, and Blood Institute; American Heart Association; World Heart Federation; International Atherosclerosis Society; and International Association for the Study of Obesity (JIS) (Table 1). In the IDF and JIS definitions, we followed the Asian population thresholds for abdominal obesity[3,4]: WC ≥90 cm for men, and ≥80 cm for women.
(Revised, revised parts are underlined): “The NCEP ATP III definition was chosen in this study because Mongolia, as an Asian country, was reported as having a higher BMI compared to Japan and China[18,19]. When using the IDF and JIS definitions, the other two widely accepted definitions of MS chosen in this study, we followed the Asian population thresholds for abdominal obesity[3,4]: WC ≥90 cm for men, and ≥80 cm for women. The NCEP ATP III, IDF and JIS definitions are different from each other in diagnostic process.
Point 8: What is considered to be an average salary per month in Mongolia? just to make it clear for our readers in context of the world.
Response 8: We thank the reviewer for this comment. We have added an information about Mongolian average salary per month in Method as follows:
(Original): Monthly income was divided into three types: upper, middle and lower, according to average salary per month.
(Revised, revised parts are underlined): “Monthly income was divided into three types: upper, middle and lower, according to average salary per month. The average monthly salary in Mongolian is about 1,500,000 Mongolian tugrik (∼450 USD).”
Point 9: Please replace the word “background” with “demographic” - characteristics.
Response 9: Thank you for your comment. We have replaced the word “background” with “demographic” in the materials and method section.
Point 10: Please provide foot note below the tables to expand abbreviations (Fr e.g., RF?)
Response 10: Thank you for your comment. We have provided foot note expanding abbreviations.
Point 11: Results: Why authors choose to report HDL-C, TG and FBG mean values in mmol/L in Table 2 when they were defined as mg/dl in Table 1. Please be consistent in reporting.
Response 11: Thank you for your important comment. This was a typo. We have revised the values in mg/dL.
Point 12: How will authors explain the discrepancy in MS prevalence according to IDF (24.8%) and JIS (26.8%) definition, when both definitions are identical for all the individual components?
Response 12: Thank you for your comment.
- For the IDF definition, WC is recognized as an essential factor even though we followed the Asian population thresholds for abdominal obesity. Furthermore, FBG is measured by ≥100mg/dl or previously diagnosed T2DM according to IDF definition. Whereas FBG is measured by ≥100mg/dl or on treatment for elevated glucose according to the IDF definition.
- In Mongolia, a previous study has reported a low prevalence of diabetes-related health knowledge among Mongolians, with as many as one in two Mongolians having never heard the term ‘diabetes’ (Demaio, A.R. et al. 2013). It is therefore uncommon to see patients in the hospital receiving diabetes diagnosis because of the low health literacy and lack of access to healthcare services. This is especially true for nomadic communities living in remote areas. The JIS definition make it easier to catch more people at risk of MS and ask them about receiving glucose treatment.
- According to our results, the prevalence of MS measured using the JIS definition was higher in both sexes than when measured using other definitions. Therefore, we recommended the JIS definition as the preferred provisional definition to identify more people at risk of MS.
Point 13: Similar questions can also be raised for the sensitivity, specificity when comparing IDF with JIS definitions and the level of agreement?
Response 13: Thank you for your comment. In Mongolia, we do not have a gold standard to measure MS. As we mentioned in Response 12, these two definitions are different from each other in diagnostic process. Therefore, we think JIS is suitable to diagnose more people at risk in Mongolia.
Point 14: The sentence discussing multivariate regression analysis is somewhat ambiguous and need to be rephrased (line 203-205)
Response 14: Thank you for your comment. We have rephrased the sentences accordingly, as follows:
(Original): After adjustment in multivariable regression analysis, older participants remained significantly (OR=1.42, 95% CI 1.01 to 1.98 in men, OR=2.63, 95% CI 1.97 to 3.51 in women). Married participants had a higher prevalence of MS only among the female population (OR=1.59, 95% CI 1.06 to 2.38).
(Revised, revised parts are underlined): “In multivariable logistic regression analysis, the 40-59 age group (aOR=1.44, 95% CI 1.01 to 2.04 in men, aOR=2.28, 95% CI 1.68 to 3.09 in women) and female 60 and over age group (aOR=3.22, 95% CI 2.04 to 5.08) in addition to married female participants (aOR=1.57, 95% CI 1.04 to 2.36) were significantly associated with MS.”
Point 15: Discussion: The paragraph discussing health literacy is irrelevant and needs to be removed, since health literacy was not assessed in this research and seems like authors conjecture at this point.
Response 15: Thank you for your comment. We have removed the health literacy part from the discussion section.
Point 16: Authors should expand their limitations considering sampling method, similarities in three definitions etc.
Response 16: Thank you for your comment. We have added the limitation about the sampling errors. Even though these three definitions have similarities, they have different ways of combining and including the components. So, we do not think the similarities of these definitions are our limitation. We have revised this section accordingly, as follows:
(Original): Several limitations of this study should be noted. First, our cross-sectional study did not allow to derive conclusions regarding causal mechanisms. Second, we did not evaluate visceral fat using computed tomography (CT) and magnetic resonance imaging (MRI) to measure central obesity. Third, our data were extracted only from the Mongolian urban population. However, sampling was carried out following the WHO STEPs Surveillance Manual, and selection bias was small.
(Revised, revised parts are underlined): “Several limitations of this study should be noted. First, our cross-sectional study did not allow to derive conclusions regarding causal mechanisms. Second, we did not evaluate visceral fat using computed tomography (CT) and magnetic resonance imaging (MRI) to measure central obesity. Third, our data were extracted only from the Mongolian urban population using multistage cluster sampling which included some sampling errors. However, our sampling was carried out following the WHO STEPs Surveillance Manual, and selection bias was smaller than non-random sampling. Besides, the current study provides the first report of a comparative study of MS prevalence in Mongolians according to three different definitions, and suggests a provisional definition for Mongolians.”

Reviewer 2 Report
In general,
The manuscript entitled "Comparison of three diagnostic definitions of metabolic syndrome and estimation of its prevalence in Mongolia" comparING the differences in MS prevalence among Mongolian urban adults based on three currently used definitions of MS that have their own features, to clarify the epidemiological situation of MS, and provide the necessary evidence for preparing diagnostic definitions.
The Instrument is well described, the Materials and Methods are clear and the Conclusions are valuable and useful for many in the field interested in the analysis.
However, it would be valuable to address the following comments:
In the MATERIAL AND METHODS include the definition of the abbreviation “STEPS” and “sop”.
Correct the ≥ in all the writing where it applies, this because it appears written in different ways in the writing for example with space and without space.
Author Response
Response to Reviewer 2 comments
In general, the manuscript entitled "comparison of three diagnostic definitions of metabolic syndrome and estimation of its prevalence in mongolia" comparing the differences in ms prevalence among mongolian urban adults based on three currently used definitions of ms that have their own features, to clarify the epidemiological situation of ms, and provide the necessary evidence for preparing diagnostic definitions. The instrument is well described, the materials and methods are clear and the conclusions are valuable and useful for many in the field interested in the analysis.
We thank the reviewer for these comments. We largely agree with the points raised and considered them in the revised version of the manuscript. Our point-by-point responses to each of the comments are as follows:
Point 1: In the material and methods include the definition of the abbreviation “steps” and “sop”.
Response 1: Thank you for your comment. We have expanded the abbreviation of SOP- Standard Operating Procedure. The STEPs survey, part of the STEPwise approach recommended by the WHO, is a survey methodology with 3-steps.
Step 1: Questionnaire survey
Step 2: Physical and anthropometric measurements
Step 3: Biochemical measurements
Point 2: Correct the ≥ in all the writing where it applies, this because it appears written in different ways in the writing for example with space and without space.
Response 2: Thank you for your important comment. We corrected all the writing of the ≥ without space. Please check the revised manuscript.

Reviewer 3 Report
This study entitled “Comparison of three diagnostic definitions of metabolic syn-2 drome and estimation of its prevalence in Mongolia” assessed the prevalence of MS in the urban population in Mongolia and suggested a superior definition. The authors reported that MS is highly prevalent among this population. The Joint Interim Statement (JIS) definition is recommended as the provisional definition. The MS is interesting and well-written. I have listed some comments below, which I hope the authors will find useful in revising the manuscript.
· Write the full term of MS, IDF…etc., at the first mention.
· It has been stated that “After a 10-min rest, the BP was measured three times, with 3-minute intervals between measurements”. Was the mean of these values calculated? Please clarify.
· Indicate the P-value in bold in the tables.
· What is the scientific justification for using 40 years as a cut-off of ages?
· In the figure, titles of the x and y axis need to be added.
Author Response
Response to Reviewer 3 comments
This study entitled “Comparison of three diagnostic definitions of metabolic syndrome and estimation of its prevalence in Mongolia” assessed the prevalence of MS in the urban population in Mongolia and suggested a superior definition. The authors reported that MS is highly prevalent among this population. The Joint Interim Statement (JIS) definition is recommended as the provisional definition. The MS is interesting and well-written. I have listed some comments below, which I hope the authors will find useful in revising the manuscript.
We thank the reviewer for these comments. We largely agree with the points raised and considered them in the revised version of the manuscript. Our point-by-point responses to each of the comments are as follows:
Point 1: Write the full term of MS, IDF…etc., at the first mention
Response 1: Thank you for your comment. We have expanded the abbreviations.
Point 2: It has been stated that “After a 10-min rest, the BP was measured three times, with 3-minute intervals between measurements”. Was the mean of these values calculated? Please clarify.
Response 2: Thank you for your important comment. Previously we used the first-time measurement’s value in our main text. According to the reviewer’s comment and international guidelines (Pickering TG et al. 2005; Kraus WE et al. 2019), we reanalyzed all the data, using the mean value of blood pressure of three times measurements instead of first-time measurement’s value and we have changed our results. Please check the revised manuscript.
Point 3: Indicate the P-value in bold in the tables.
Response 3: Thank you for your comment. We have indicated the P-value in bold in the tables.
Point 4: What is the scientific justification for using 40 years as a cut-off of ages?
Response 4: Thank you for your comment. We have changed the age group into 3 categories: young adults (<40), middle adults (40-59), and old adults (60 and over). (Chovalopoulou ME et al. 2017) So, we reanalyzed and revised the results in main text. Please check the revised manuscript.
Point 5: In the figure, titles of the x and y axis need to be added.
Response 5: Thank you for your comment. We have added titles of the x and y axis in the figure.

Round 2
Reviewer 1 Report
Dear Authors,
Thank you for revising the manuscript to address my comments. The manuscript is now significantly improved. I have no further comments.